# All-Optically Controlled Terahertz Modulation by Silicon-Grown CdSe/CdZnS Colloidal Quantum Wells

**DOI:** 10.3390/nano15201597

**Published:** 2025-10-20

**Authors:** Reyihanguli Tudi, Zhongxin Zhang, Xintian Song, Bumaliya Abulimiti, Mei Xiang

**Affiliations:** Xinjiang Key Laboratory for Luminescence Minerals and Optical Functional Materials, School of Physics and Electronic Engineering, Xinjiang Normal University, Urumqi 830054, China; rayhan2024@163.com (R.T.); sxtgtx@126.com (X.S.);

**Keywords:** CdSe/CdZnS CQWs, THz modulator, all-optically controlled, heterojunction

## Abstract

The CdSe/CdZnS colloidal quantum wells, with their exceptionally high carrier mobility and ultrafast response characteristics, emerge as highly promising candidate material for high-performance active terahertz modulators—indispensable core components critical for next-generation communication technologies. A high-performance, cost-effective terahertz modulator was fabricated through spin-coating CdSe(4ML)/CdZnS nanosheets onto a silicon substrate. This all-optical device demonstrates broadband modulation capabilities (0.25–1.4 THz), achieving a remarkable modulation depth of 87.6% at a low power density of 2 W/cm^2.^ Demonstrating pump-power-efficient terahertz modulation characteristics, this core–shell composite shows immediate applicability in terahertz communication systems and non-destructive testing equipment.

## 1. Introduction

In the era of information explosion and rapid technological iteration, terahertz waves, with their sub-picosecond pulse width, low photon energy, and specific spectral responses to various substances, have become strategic resources in fields such as information transmission [1,2], biomedical diagnosis [3], and security detection [4]. In addition, Terahertz waves have found a variety of new applications, such as wireless communication, biological sensing, social security, medical diagnostics, detecting counterfeit pharmaceuticals, and fundamental research, due to their high spectral resolution, broad bandwidth, better penetration depth, low photo energy, and high transparency [4,5,6,7,8]. However, the development of terahertz communication, a pivotal enabler for 6G, is currently hindered by the absence of high-performance, dynamically reconfigurable, and monolithically integrable modulators. Terahertz time-domain spectroscopy is an important experimental method for studying the ultrafast dynamics and photo-conductive characteristics of charge carriers in materials [9,10]. Terahertz time-domain spectroscopy technology is a coherent detection technique that has the advantage of being able to detect the optical properties of materials in a non-contact and non-destructive manner, including transmissive and reflective Terahertz time-domain spectroscopy techniques [11]. The all-optically controlled THz modulator represents a revolutionary leap in THz photonics [12,13]. By circumventing the limitations imposed by electrical constraints, this cutting–edge device provides inherent benefits such as increased modulation depth, wider operational bandwidth, ultrafast switching speeds, and superior sensitivity [14]. These characteristics make it an indispensable element in the development of advanced THz systems. For instance, a graphene-based THz modulator achieved nearly perfect amplitude modulation with a 99.9% modulation depth and a bandwidth ranging from 0.25 to 1.6 THz by tuning the graphene’s conductivity via an electrical gate [15]. Spin-coating is a process in which a solution containing the precursor materials (typically metal salts or metal–organic compounds) is dispensed onto a substrate and spread into a uniform liquid film by high-speed rotation. The key point is that this liquid film itself is not epitaxial. The epitaxial structure is formed during the subsequent heat treatment (annealing) process. It is entirely possible to prepare epitaxial single-crystal thin films with lattice-order continuity across the interface using the spin-coating method. In high-speed terahertz communication systems, heterojunction terahertz modulators can achieve fast modulation of terahertz waves, improving communication speed and signal quality. Constructing terahertz detectors using the characteristics of heterojunctions can improve the sensitivity and response speed of the detectors, achieving efficient terahertz signal reception. For example, the NiO/Si heterojunction capitalizes on the strong p-n junction characteristics and defect-mediated photoconductivity at the interface. Under optical excitation, the photo-generated carriers at the NiO/Si interface exhibit ultrafast dynamics, enabling efficient modulation of THz waves via transient conductivity changes [16]. Meanwhile, the CdS_NWs_/Si heterojunction exploits the unique 1D–3D hybrid structure, where the high aspect ratio of CdS nanowires enhances light absorption and carrier transport, leading to broadband THz modulation [17]. Select materials with different terahertz response characteristics to construct heterojunctions, such as combining semiconductor materials with high carrier mobility and materials with strong absorption or scattering effects on terahertz waves. While traditional materials demonstrate inherently weak interactions with terahertz waves, state-of-the-art colloidal quantum wells (CQWs) are hampered by insufficient stability and integration capabilities.

Semiconductor colloidal quantum wells (CQWs), also referred to as nanoplatelets (NPLs), represent an emerging class of nanomaterials with atomic-level precision in thickness control, garnering significant attention in the field of optoelectronics [18,19]. As an important subclass of nanocrystals (NCs), CQWs exhibit remarkable optical and electronic properties stemming from their strong one-dimensional exciton confinement and tunable heterostructure design [20]. These materials demonstrate unique advantages, including large absorption cross-sections [21], narrow and tunable spontaneous emission [22], and pronounced amplified spontaneous emission. Such exceptional characteristics position CQWs as highly promising candidates for next-generation optoelectronic devices, such as lasers, light-emitting diodes (LEDs), and photo detectors [23]. CdSe/CdZnS Colloidal Quantum Wells (CQWs) represent an important class of core/shell semiconductor nanostructures that combine the strong quantum confinement of CdSe nanoplatelets with the enhanced stability and tunable optoelectronic properties provided by the CdZnS alloy shell. CdSe/CdZnS colloidal quantum wells (CQWs) exhibit remarkable potential as dynamically tunable materials for terahertz (THz) wave modulation, owing to their unique quantum confinement effects and adjustable optoelectronic properties [24,25]. Through external optical excitation or electric field modulation, the carrier concentration and band structure in quantum wells can undergo rapid modifications [26], thereby influencing their absorption, reflection, or transmission characteristics for THz waves. For instance, photoexcitation can induce carrier transitions within the quantum wells, altering their dielectric constant and consequently modulating the propagation behavior of THz waves [27]. This paper introduces a novel design based on CdSe/CdZnS core/shell CQWs synthesized directly on silicon substrates. This approach capitalizes on the distinctive optoelectronic characteristics of the CQWs and their innate CMOS compatibility, aiming to bridge the critical gap in the realization of advanced terahertz modulation devices. For probing carrier dynamics in heterointerfaces, terahertz wave modulated optical responses of CdSe/CdZnS colloidal quantum well heterostructures spin-coated on high-resistance silicon were systematically characterized via time-domain spectroscopy synchronized with a 532 nm CW excitation source. The novel material exhibits outstanding performance in broadband terahertz wave modulation at 0.83 THz, achieving a remarkably high modulation depth of approximately 87.6% under a pump laser power density of 2 W/cm^2^. We have experimentally and theoretically demonstrated that it is highly efficient modulation for THz waves is due to the photon-generated carrier effect on the CdSe/CdZnS–Si interface, which paves a promising route for designing novel all-optical micro-functional devices.

## 2. Characterization of Sample and Experiment Setup

The bare silicon and sapphire substrates were ultrasonically cleaned in acetone and ethanol, sequentially, for 30 min each. Subsequently, a CdSe(4ML)/CdZnS core/shell colloidal quantum well (150 μL) was deposited onto the substrate via drop coating and placed under vacuum until all solvents had completely evaporated. Then, the solution was spin-coated onto an ultrasonically cleaned silicon substrate (diameter = 500 µm, resistivity = 5000 Ω cm^−2^), rotated at a speed of 1000 rpm for 30 s, and annealed at 120 °C for 30 min to obtain sample 1 (Sam1) (CdSe/CdZS/Si). Under the same conditions as Sam1, sample 2 (Sam2) was prepared by rotating at a speed of 1000 rpm. Furthermore, CdSe/CdZnS structures were prepared on sapphire substrates (thickness = 500 microns) using the same method.

In this study, we used a self-made terahertz time-domain spectrometer to measure transmission spectra over a broad frequency range of 0.3 to 1.4 THz, providing detailed insights into the optical properties of the sample. A path illuminates the photo-conductive antenna (PCA-40-05-10-800-h, Bartop, Stuttgart, Germany) to generate terahertz pulses, while another path is delayed and illuminates the detector antenna (BPCA-100-05-10-800-h, Bartop, Germany) for terahertz detection. The generated terahertz beam was focused on the sample. Linear translation stages were used to precisely control the delay between femtosecond pulses. By scanning the delay time, the system recorded the time-domain terahertz spectrum. A Fourier transform was performed on these spectra to obtain amplitude and frequency information. Figure 1a presents the experimental layout of the THz-TDS. For transmission measurements. Using a femtosecond laser (Spectra-Physics, Mai Tai, Milpitas, CA, USA) with a center wavelength of 800 nm as the optical pump source. The transmitted THz light was then detected, and the signal was acquired and amplified using a lock-in amplifier. The pulse has a duration of 100 fs and a repetition rate of 84 MHz. The 50:50 beam splitter splits the femtosecond laser beam into two paths. The spot diameters of terahertz beams were 1.8 mm, with a maximum power of 2 W/cm^2^ was used to excite the sample, and the schematic of the optically controlled terahertz modulator is represented in Figure 1b. The transmission spectrum of air under the same conditions is taken with reference to the terahertz transmission spectra of CdSe/CdZnS/Si, CdSe/CdZnS/Sapphire, and bare silicon were measured at different laser power densities.

## 3. Result and Discussion

### 3.1. Characterization of CdSe/CdZnS Core/Shell

Figure 2a,b show the surface morphology of thin films was characterized using scanning electron microscopy (SEM, Hitachi S-4800, Hitachi, Tokyo, Japan). In the image, the white area on the left represents the edge of the substrate after spin-coating, while the region within the yellow frame corresponds to the spin-coated surface. From the surface view as Figure 2a, the CdSe/CdZnS layer on the Si substrate exhibits a relatively uniform and smooth morphology, suggesting good adhesion and film-forming properties during deposition.

Figure 2b shows the cross-sectional SEM imaging of the multilayer structure of CdSe/CdZnS/Si. The interface of the spin-coated material is indicated by the arrowed portion above the yellow line, which often implies a high-quality growth process and good lattice matching. High-quality interfaces are crucial for reducing interface state density and improving carrier efficiency, especially in optoelectronic applications. Figure 2c,d shows the Tauc and Absorption with PL centered spectra of the UV-Vis absorption spectra of the CQW solution, respectively. The energy gap Eg = 2.5 eV was calculated through UPS measurement and band theory calculation. This means that it has good responsiveness to light in the visible light region. The PL of the CQWs exhibits a peak at 644 nm, accompanied by a full-width at half-maximum (FWHM) of 16 nm. As shown in Figure 2d, the position of the fluorescence emission peak is slightly higher than that of the absorption peak, which is due to the Stokes shift phenomenon; the energy of fluorescence emission is slightly lower than the energy absorbed.

### 3.2. Terahertz Transmission Results

The transmission spectra of CdSe/CdZnS core/shell were measured using a self-made THZ time-domain spectroscopy (THZ-TDS) system at different power densities under a 532 nm CW pumped laser. The relevant results are presented in Figure 3. From Figure 3a,b, it can be seen that as the pump power density gradually increases from 0 to 2 W/cm^2^, the amplitude of terahertz waves continues to decrease. This phenomenon clearly indicates that both Sam1 and Sam2 can significantly modulate terahertz waves. Further comparison reveals that under the same pump power conditions, Sam1 generally outperforms Sam2 in modulating terahertz waves. The numerical value of terahertz transmittance can be directly used as a key indicator to measure the terahertz wave transmission characteristics of materials. In addition, the data in Figure 3c shows that even with a high laser intensity of 2 W/cm^2^, the modulation ability of bare silicon substrates for terahertz waves is still weak. the acquisition of amplitude transmission spectrum needs to be based on THz TDS measurement data and combined with Equation (1) to calculate. In the equation used, $Εr\left(\omega \right)\;$ and $Εs\left(\omega \right)$ represent the Fourier transform of the air reference signal and the sample signals (with or without 532 nm CW laser pumping). As shown in Figure 3d–f, there is no significant difference in the terahertz transmission curves of Sam1, Sam2, and bare silicon when there is no pump laser irradiation, indicating that the difference in terahertz transmission performance among the three is small when they are not excited by light. As the pump power density gradually increases, the terahertz transmittance of all three samples shows a significant decrease trend. From a physical mechanism perspective, the pumped laser irradiation on a silicon substrate triggers the generation of photo-generated charge carriers. The appearance and concentration changes in charge carriers disrupt the original conductivity balance of the silicon substrate, hinder the effective transmission of terahertz waves, and, ultimately, lead to a decrease in transmittance. The Fabry–Perot effect can cause fluctuations in transmission spectra, and the formation of this effect is related to two factors: one is the reflection of terahertz waves at the boundary between the sample and air, and the other is the multiple reflections of terahertz waves inside the sample [28]. When the pump power density is kept consistent and the transmittance of Sam1 and Sam2 is compared, it can be found that the terahertz transmittance value of Sam1 is significantly lower than that of Sam2, reflecting that under the same light excitation conditions, Sam1 has a better modulation effect on terahertz waves. Stronger signal contrast is an important prerequisite for improving detection sensitivity and data transmission efficiency, which relies on high-performance terahertz modulators with higher modulation depth.(1)$$\left[T(\omega )=|(E_{s}\left(\omega \right))|∕|E\_r\;(\omega )|\right]$$

In view of this, to visually evaluate the differences in optical terahertz modulation performance between Sam1, Sam2, and bare silicon, the study selected 0.25–1.4 THz as the analysis frequency band and calculated the modulation depth of the three samples. In this study, the modulation depth is defined as follows:(2)$$Μ=\frac{{Τ}_{pump}-{Τ}_{off}}{{Τ}_{off}}$$
where Toff represents the THz wave transmission without laser power, while Tpump represents the THz wave transmission with pumping laser power.

Figure 4a–c demonstrate broadband modulation for all samples (Sam1, Sam2, and bare silicon). However, the maximum modulation depths of Sam1 and Sam2 are significantly higher compared to bare silicon. Figure 4d shows the dependence of the modulation depth at 0.8 THz on the laser power density. All samples exhibit a gradual increase in modulation depth with increasing optical power density, though the introduction of this layer may facilitate the generation of photogenerated carriers (electron-hole pairs) at the interface after irradiation. The increase in carrier concentration plays a key role in enhancing the modulation depth by affecting the electrical conductivity and optical response of the material. This dynamic mechanism allows the material to effectively respond to varying laser powers, thereby achieving modulation of terahertz wave transmission characteristics. To shed light on the modulation mechanism of the films, the terahertz time-domain spectra of bare sapphire and Sam3 were investigated under identical conditions. 

Figure 5a,b illustrate that the terahertz transmission of bare sapphire and Sam3 (CdSe/CdZnS/Sapphire) remains unchanged as the continuous laser power density is gradually increased from 0 to 2.0 W/cm^2^. These results suggest that CW laser excitation at 532 nm on an insulating sapphire substrate does not contribute to the generation of charge carriers at this particular wavelength. As a result, the terahertz absorption of CdSe/CdZnS films is negligible. Under illuminated conditions, the CdSe/CdZnS/Si bilayer structure exhibits significant attenuation in terahertz wave transmission intensity.

This phenomenon arises from the strong built-in electric field formed at the heterojunction interface, which effectively drives the separation of photogenerated carriers and extends their recombination lifetime. This characteristic of interfacial charge separation has been confirmed as a key factor in regulating terahertz wave transmission, providing a foundation for efficient terahertz modulation. Understanding the THz wave modulation mechanism in Si-grown CdSe/CdZnS CQW requires an analysis of the carrier properties. To this end, Sam1 was employed to analyze the modulation process. Figure 6a,b shows the changes in the refractive index and conductivity of Sam1 at different pump powers, respectively. When the laser intensity increased to 2 W/cm^2^, the refractive index of Sam1, for example, at 0.25–1.4 THz, had an apparent decrease from 3.43 to 2.97, as shown in Figure 6. Different optical power densities significantly affect the refractive index characteristics of materials.

This is likely due to changes in the distribution of electrons within the material when light interacts with it, or it may also result from changes in conductivity associated with carrier density. Thus, we also calculated the conductivity, as shown in Figure 6b. Under 2 W/m^2^ of laser light, the conductivity σ_sub_ of the CdSe/CdZnS/Si reached 3626 s/m at 0.83 THz. This means that the carrier density of Si-grown CdSe/CdZnS nanosheets had a much larger increase under the weak pumping power. In the CdZnS shell, the incorporation of Zn alters the crystal structure and electron cloud distribution, which may lead to increased carrier scattering. At the same time, Si is used as a substrate material, and its internal lattice defects and impurities will also hinder the movement of carriers, reduce the carrier mobility, and then reduce the overall conductivity of the material. Therefore, deeply understanding and precisely controlling the carrier transfer process at this interface is crucial for optimizing the optoelectronic performance of such heterostructures. The ensuing discussion of carrier separation processes at the CdSe/CdZnS/Si interface serves to substantiate our claims and delineate the charge transfer mechanism under CW excitation. The energy band alignment of the CdSe/CdZnS/Si heterostructures before and after contact is shown in Figure 7a and Figure 7b, respectively. Band engineering is achieved by establishing a heterojunction between CdSe/CdZnS and n-type Si.

Upon photoexcitation, the generated carriers are separated by the built-in electric field: photogenerated electrons in the CdSe/CdZnS migrate toward the n-type Si under field-driven transport, while holes migrate in the opposite direction. This spatial separation mechanism effectively reduces the probability of carrier recombination, thereby significantly enhancing carrier collection efficiency. The vacuum energy level was 0 eV. Since the values of bandgap (Eg2) and work function (ϕ2) for intrinsic silicon were well-known (1.12 eV and 4.6 eV), we estimated the values of Ec2, Ev2, and E_F_2 to be 4.04 eV, 5.16 eV, and 4.6 eV, respectively. The energy level parameter values of CdSe/CdZnS colloidal quantum wells and Si are shown in Table 1. The ultraviolet photoelectron spectra (UPS) of CdSe/CdZnS were performed to confirm the transfer mechanism of photogenerated electrons. As shown in Figure 7c,d, the valence band maximum (EVM) of CdSe/CdZnS was 4.0 eV, and the secondary electron cutoff (Ecut off) of pure CdSe/CdZnS was recorded at 17.2 eV. The work function (ϕ) calculated by the difference between the photon energy (21.22 eV) and Ecut off was 4.0 eV for CdSe/CdZnS.

## 4. Conclusions

A high-sensitivity all-optical terahertz modulator based on CdSe(4ML)/CdZnS/Si heterojunction was prepared using spin coating technology. The transmission spectra and modulation response were measured using a homemade terahertz time-domain spectrometer. The broadband modulation performance of CdSe/CdZnS/Si was observed in the range of 0.25–1.4 THz using a 532 nm CW laser, achieving a remarkable modulation depth of 87.6% at a low power density of 2 W/cm^2^. High-performance modulators typically seek higher modulation depths to provide stronger signal contrast, thereby improving detection sensitivity or data transmission efficiency. By comparing Sam1 and Sam2, it was found that there are differences in the terahertz modulation performance of the heterojunction formed at different rotational speeds, while the significant enhancement of the optoelectronic performance caused by the CdSe/CdZnS CQW layer further improves the modulation efficiency. Therefore, PL spectra show that the CdSe/CdZnS quantum well has a luminescence peak at 644 nm (Eg = 2.5 eV) with an FWHM of 16 nm, indicating that the material possesses high optical quality, narrow energy band distribution, and good size uniformity. A small FWHM suggests that the carrier energy states are concentrated, which is conducive to efficient photoelectric response and narrow linewidth emission, indicating potential applications in terahertz modulators, LEDs, or lasers. Further analysis of the carrier separation process at the CdSe/CdZnS/Si interface helps to validate our view and elucidate the charge transfer mechanism under continuous wave (CW) excitation. Experiments have found that the conductivity of this structure is relatively low. However, despite the low conductivity, the CdSe/CdZnS/Si heterojunction can achieve a high modulation depth in the terahertz band due to its unique material properties and optimized structural design. This CdSe/CdZnS quantum well material, based on the silicon substrate, exhibits higher efficiency and wider bandwidth in terahertz wave modulators, and can be easily integrated into terahertz silicon-based waveguides, chips, and metamaterial structures, providing a broader and promising application space for the development of terahertz technology.

## Figures and Tables

**Figure 1 nanomaterials-15-01597-f001:**
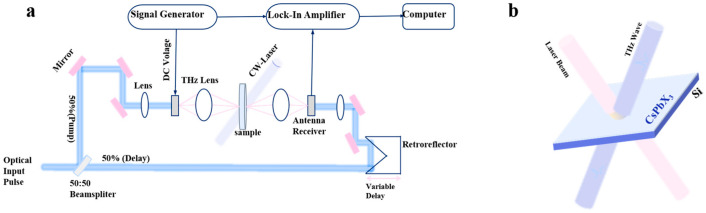
(**a**) The schematic diagram of the experimental setup of THz-TDS combined with a modulating laser. (**b**) The schematic diagram of an optically controlled THz modulator.

**Figure 2 nanomaterials-15-01597-f002:**
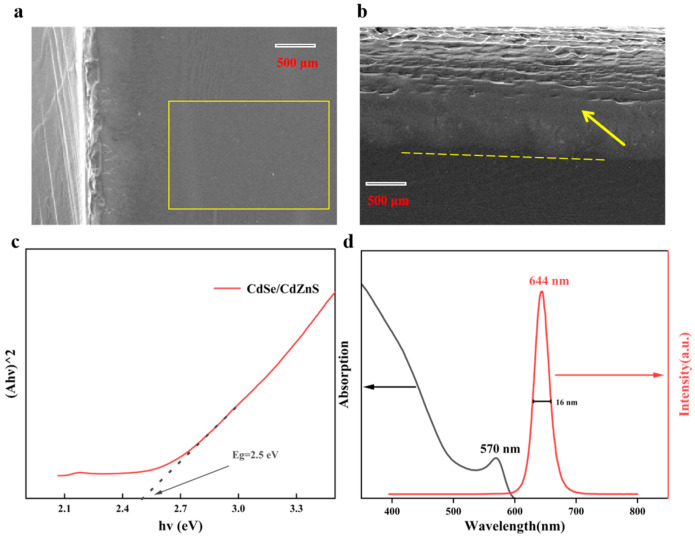
SEM images: (**a**) CdSe/CdZnS/Si film and (**b**) cross-section of CdSe/CdZnS/Si film. (**c**) Tauc plots (αhν)^1/2^ versus hν) give the band gap of CdSe/CdZnS core/shell CQWs (the solid line and dotted line represent the absorption edge and tangent line). (**d**) Absorption and PL centered spectra of CdSe/CdZnS core/shell CQWs.

**Figure 3 nanomaterials-15-01597-f003:**
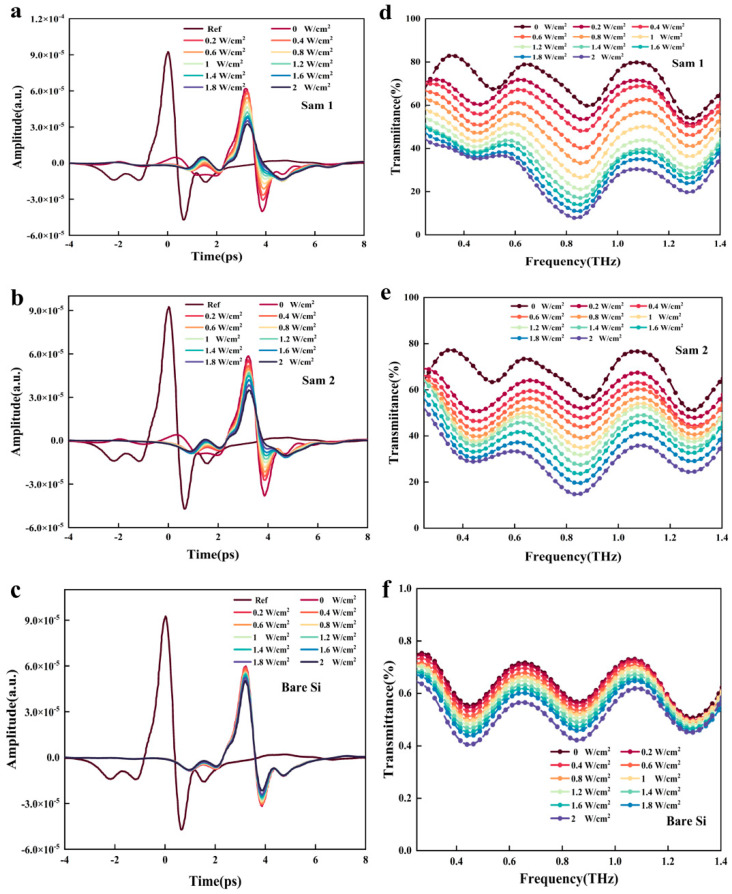
THz time-domain signals of (**a**) Sam1, (**b**) Sam2, and (**c**) bare Si, and transmission spectra of (**d**) Sam1, (**e**) Sam2, and (**f**) bare Si under different pumping power.

**Figure 4 nanomaterials-15-01597-f004:**
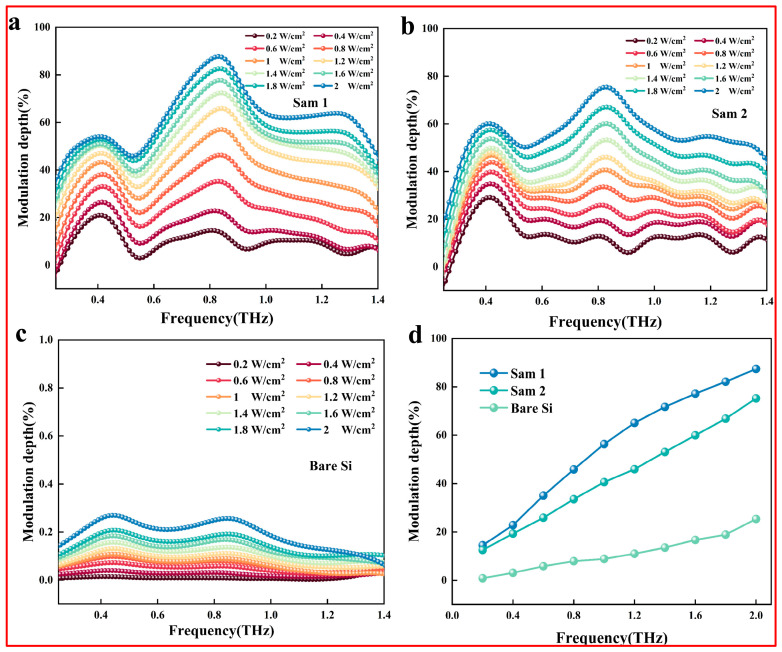
Modulation depth of samples at (**a**) Sam1, (**b**) Sam2, and (**c**) bare Si under different pump intensities. (**d**) Modulation depths of the bare Si, Sam1, and Sam2 as a function of the pump intensity at 0.8 THz.

**Figure 5 nanomaterials-15-01597-f005:**
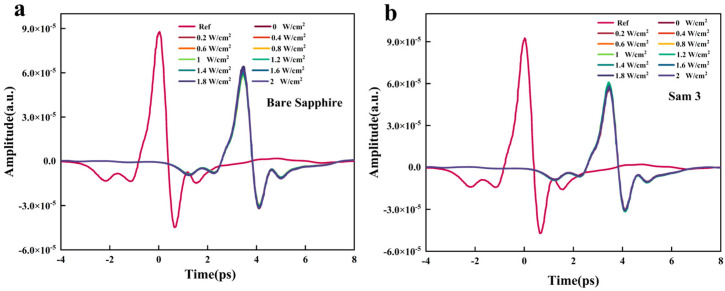
THz time-domain transmission of (**a**) sapphire and (**b**) Sam3 with the CW pump intensity ranging from 0 to 2 W/cm^2^.

**Figure 6 nanomaterials-15-01597-f006:**
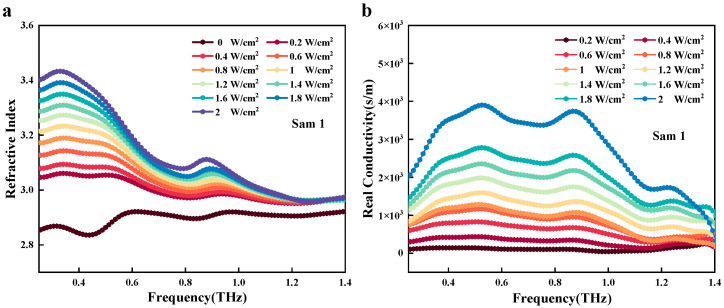
(**a**) Refractive index and (**b**) Real conductivity of Sam1 with the CW pump intensity ranging from 0 to 2 W/cm^2^.

**Figure 7 nanomaterials-15-01597-f007:**
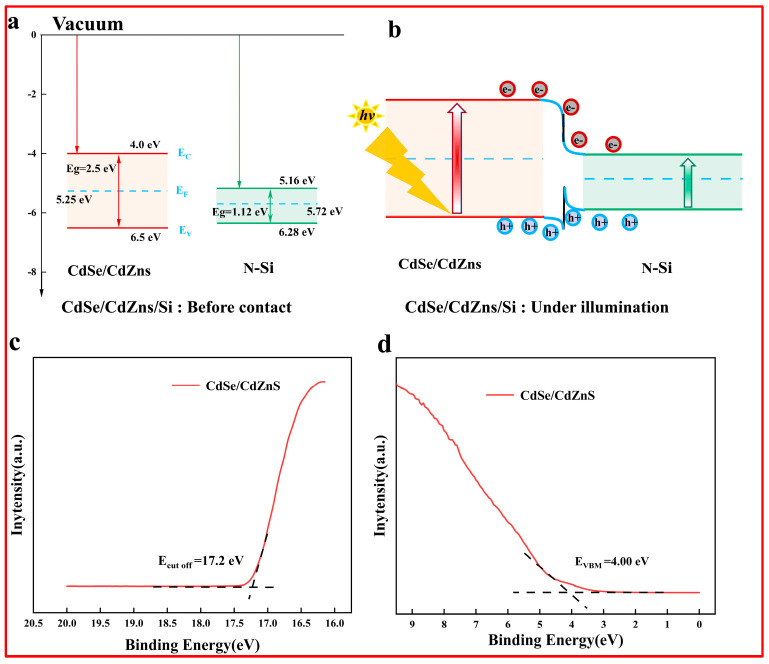
(**a**) Energy band structure of CdSe/CdZnS and Si before contact. (**b**) Band alignment of the CdSe/CdZnS/Si heterostructure under laser illumination and schematic diagram of the charge carriers in CdSe/CdZnS/Si samples. The UPS of (**c**) secondary edge region and (**d**) low energy onset region of CdSe/CdZnS CQWs.

**Table 1 nanomaterials-15-01597-t001:** Energy level parameters (conduction band minimum, valence band maximum, band gap, and Fermi level value) for CsPbCl_3_, CsPbBr_3_, and Si.

E	CdSe/CdZnS	Si
E_CB_	6.5 eV	6.28 eV
E_VB_	4.0 eV	5.16 eV
E_g_	2.5 eV	1.12 eV
E_F_	5.25 eV	5.72 eV

## Data Availability

The original contributions presented in this study are included in the article. Further inquiries can be directed to the corresponding authors.

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
