# Peer review of "All-Optically Controlled Terahertz Modulation by Silicon-Grown CdSe/CdZnS Colloidal Quantum Wells"

_nanomaterials, 2025, doi:10.3390/nano15201597_

Round 1
Reviewer 1 Report
Comments and Suggestions for Authors
This paper reports experimental results on modulation of THz transmission by colloidal quantum wells using pulsed laser stimulation. Colloidal quantum wells and their applications are interesting. THz modulators may have value. The paper presents original data with reasonable interpretation and should be published. I have just a few comments for the authors to consider. There is no need in Figure 1 for artistic renderings of optics mounts and instrumentation boxes. This actually makes it difficult to see the important function. It would be better to make a simplified 2D line drawing where the nature of each component is more clear without any extraneous information. The current Fig. 1 would be ok for a talk.
Section 2: What is an ultrasonic substrate? I think they mean ultrasonically-CLEANED substrate.
Section 2: What is the unit "rpm/S"? I think they mean "rpm".
Fig. 2 a) Contrast is so bad that nothing can be seen at all, so this information contains no information. In b) is entire gray part the edge, or are we seeing some of the surface? Better to indicate with arrows the layers that are supposed to be obvious.
Subsequent paragraph mentions good "lattice-matching". Not being very familiar with these materials, is it really possible that a spin-on coating can make crystalline epitaxial layers with lattice order continuing across the hetero interface from one material to the other? If this is true and has been supported by cross-sectional TEM measurements, the authors might consider asserting it to be so in the introduction.
Author Response
Reviewer's Comments:1) There is no need in Figure 1 for artistic renderings of optics mounts and instrumentation boxes. This actually makes it difficult to see the important function. It would be better to make a simplified 2D line drawing where the nature of each component is more clear without any extraneous information.
Our reply: Again, we sincerely appreciate your professional and positive comments. We sincerely thank you very much for pointing this out. As you nicely suggested, we have remade the 2D line diagram. And we hope this revision could be acceptable for you. Please check the abstract for details. Again, thank you very much for your useful suggestions.
Reviewer's Comments:2) What is an ultrasonic substrate? I think they mean ultrasonically-CLEANED substrate.
Our reply: We are sorry for our mistake. As the reviewer pointed out, it is an ultrasonically-cleaned substrate.
Reviewer's Comments:3) What is the unit "rpm/S"? I think they mean "rpm".
Our reply: As the reviewer pointed out. It's not "rpm/S ", it's "rpm". It was our mistake in expression.
Reviewer's Comments:4) Fig. 2a) Contrast is so bad that nothing can be seen at all, so this information contains no information. In b) is entire gray part the edge, or are we seeing some of the surface? Better to indicate with arrows the layers that are supposed to be obvious.
Our reply: Figure 2a aims to show the uniformity of the sample spin-coated on the silicon substrate, so the surface is relatively smooth. Figure 2b is indeed a partial surface. We will revise it according to the reviewer's suggestion and add arrows in the layer to indicate the interface.
Reviewer's Comments:5) Subsequent paragraph mentions good "lattice-matching". Not being very familiar with these materials, is it really possible that a spin-on coating can make crystalline epitaxial layers with lattice order continuing across the hetero interface from one material to the other? If this is true and has been supported by cross-sectional TEM measurements, the authors might consider asserting it to be so in the introduction.
Our reply: Yes, it is entirely possible to prepare epitaxial single-crystal thin films with lattice-order continuity across the interface using the spin-coating method. We have added explanatory notes in the introduction according to the reviewers' suggestions.
And finally, most of all, thank you again for your positive comments and valuable suggestions to improve the quality of our manuscript.
Reviewer 2 Report
Comments and Suggestions for Authors
The manuscript investigates the Si-based CdSe/CdZnS colloidal quantum wells for THz modulation. The device is claimed to achieve high performance while remaining cost-effective. Particularly, remarkable modulation depth of 87.6% at a low power density of 2W/cm2 is reported. I recommend revision before the paper is accepted for publication.
- Clear signs of AI/LLM assisted text generation is seen in the manuscript. According to MDPI rules, the use of AI/LLM must be explicitly declared.
- In the introduction, the paper motivation is missing. What limitations exist for CQW, what is missing in the state of the art study, and how this works helps to address such gap.
- For device fabrication, schematic of fabrication process or flow chart is recommended.
- “rotated at a speed of 1000 rpm/S for 30 seconds” unit is rpm/s?
- For preparation of SAM 2, is RPM 1000 rpm applied for 30 seconds (so sam 1 is processed the same as SAM2)?
- More SEM image of prepared sample should be shown to demonstrate quality of fabricated device, e.g. surface condition, layer separation, edge condition, etc.
- All equations must be indexed.
- For the definition of modulation depth, provide reference if exists.
- “This is likely due to changes in the distribution of electrons within the material when light interacts with it, or it may also result from changes in conductivity associated with carrier density” how to demonstrate this hypothesis? Do you have measurement of the conductivity?
- “Ecut off was 4.02 eV for CdSe/CdZnS” also for table 1, provide reference support.
- Discuss how the proposed design can inspire other relevant field.
Round 2
Reviewer 2 Report
Comments and Suggestions for Authors
I suggest acceptance.